# Targeted Polymer-Based Probes for Fluorescence Guided Visualization and Potential Surgery of EGFR-Positive Head-and-Neck Tumors

**DOI:** 10.3390/pharmaceutics12010031

**Published:** 2020-01-01

**Authors:** Robert Pola, Eliška Böhmová, Marcela Filipová, Michal Pechar, Jan Pankrác, David Větvička, Tomáš Olejár, Martina Kabešová, Pavla Poučková, Luděk Šefc, Michal Zábrodský, Olga Janoušková, Jan Bouček, Tomáš Etrych

**Affiliations:** 1Institute of Macromolecular Chemistry, Czech Academy of Sciences, Heyrovského sq. 2, 162 06 Prague 6, Czech Republic; pola@imc.cas.cz (R.P.); bohmova@imc.cas.cz (E.B.); filipova@imc.cas.cz (M.F.); pechar@imc.cas.cz (M.P.); janouskova@imc.cas.cz (O.J.); 2Center for Advanced Preclinical Imaging (CAPI), First Faculty of Medicine, Charles University, U Nemocnice 5, 120 00 Prague 2, Czech Republic; Jan.Pankrac@lf1.cuni.cz (J.P.); sefc@cesnet.cz (L.Š.); 3Institute of Biophysics and Informatics, First Faculty of Medicine, Charles University, Salmovská 1, 120 00 Prague 2, Czech Republic; david.vetvicka@gmail.com (D.V.); tomas.olejar@seznam.cz (T.O.); martina.kabesova@lf1.cuni.cz (M.K.); pavla.pouckova@lf1.cuni.cz (P.P.); 4Department of Otorhinolaryngology Head and Neck Surgery, First Faculty of Medicine, Charles University and University Hospital Motol, V Úvalu 84, 150 06 Prague 5, Czech Republic; michal.zabrodsky@fnmotol.cz

**Keywords:** fluorescence, polymeric conjugate, guided surgery, HPMA, tumor, Head and Neck carcinoma

## Abstract

This report describes the design, synthesis and evaluation of tumor-targeted polymer probes to visualize epidermal growth factor receptor (EGFR)-positive malignant tumors for successful resection via fluorescence guided endoscopic surgery. Fluorescent polymer probes of various molecular weights enabling passive accumulation in tumors via enhanced permeability and retention were prepared and evaluated, showing an optimal molecular weight of 200,000 g/mol for passive tumor targeting. Moreover, poly(*N*-(2-hydroxypropyl)methacrylamide)-based copolymers labeled with fluorescent dyes were targeted with the EGFR-binding oligopeptide GE-11 (YHWYGYTPQNVI), human EGF or anti-EGFR monoclonal antibody cetuximab were all able to actively target the surface of EGFR-positive tumor cells. Nanoprobes targeted with GE-11 and cetuximab showed the best targeting profile but differed in their tumor accumulation kinetics. Cetuximab increased tumor accumulation after 15 min, whereas GE 11 needed at least 4 h. Interestingly, after 4 h, there were no significant differences in tumor targeting, indicating the potential of oligopeptide targeting for fluorescence-navigated surgery. In conclusion, fluorescent polymer probes targeted by oligopeptide GE-11 or whole antibody are excellent tools for surgical navigation during oncological surgery of head and neck squamous cell carcinoma, due to their relatively simple design, synthesis and cost, as well as optimal pharmacokinetics and accumulation in tumors.

## 1. Introduction

Surgery remains the primary treatment option for most solid malignant tumors [1]. Precise and complete resection of the whole tumor without unnecessary removal of the neighboring healthy tissue is a prerequisite for a successful outcome [2]. Unfortunately, visual distinction between the malignant and healthy tissue using only the surgeon’s naked eye is often almost impossible. Therefore, development of a tumor-specific marker that would visualize the tumor margins is highly desirable.

Radical resection with adequate negative margins is one of the most important factors influencing the prognoses of patients with head and neck squamous cell carcinoma (HNSCC). Traditionally, the safe margin is defined as greater than 5.0 mm, but similar outcomes are possible with a surgical margin of more than 2.2–3.0 mm [3,4].

Optical imaging could be used for precise identification of the tumor margins. Indeed, narrow-band imaging (NBI) is already used in clinical practice. This horizontal imaging method is based on the contrast between pathological and healthy microvascular architecture [5]. For evaluation of the deeper tissue layers, vertical diagnostic methods, such as confocal or high resolution endomicroscopy [6], could be used.

Recently, a novel strategy to highlight tumor tissue, mainly tumor margins, using actively and passively targeted fluorescent nanoprobes, was described [7]. Intraoperatively, fluorescence intensity could guide the surgeon to resect tumor tissue together with a safe margin. Epidermal growth factor receptor (EGFR) is known to be overexpressed on the cell surface of various types of malignant tumors, including breast and lung adenocarcinomas (40% of cases) [8,9], anal cancers [10], glioblastoma (50%) [11] and epithelial tumors of the head and neck (80–100%) [12]. Recently, we reported *N*-(2-hydroxypropyl)methacrylamide (HPMA)-based polymer probes for fluorescence-guided surgery targeted to EGFR with the oligopeptide YHWYGYTPQNVI (GE-11) [13]. It was demonstrated that these EGFR-targeted polymer nanoprobes accumulated to a higher extent in EGFR-positive cells in vitro when compared to the non-targeted control polymer. Similarly, EGFR-targeted nanoprobes were significantly accumulated in EGFR-positive tumors in vivo, thus showing the future potential of these polymer nanoprobes within fluorescence-navigated surgery. Moreover, the EGFR-targeted polymer nanoprobe demonstrated a significant signal in the margin of the tumors, thus clearly showing the benefit of the nanoprobe for the precise navigation of surgeons during solid-tumor removal.

The present study compared several HPMA-based fluorescent probes for EGFR-specific labeling of hypopharyngeal carcinoma cells (FaDu) and breast adenocarcinoma cells (MDA-MB-231) [14,15]. Besides the already mentioned GE-11 targeting oligopeptide designed, prepared and evaluated, polymer probes were also targeted with human epidermal growth factor (EGF), which is the natural ligand for EGFR, and with a monoclonal antibody cetuximab, which is a clinically approved EGFR inhibitor distributed under the trade name Erbitux. EGFR-specific cell binding of all the targeted polymer probes in vitro was evaluated using flow cytometry; fluorescent visualization of EGFR-positive tumors in vivo was performed using an in vivo imaging system in-Vivo Xtreme (Bruker).

In parallel with the actively targeted polymer probes, we also investigated high molecular weight fluorescent polymers without the targeting ligands described above. These polymers passively accumulate within the tumor tissue due to the so-called enhanced permeability and retention (EPR) effect [16,17], which is based on a leaky neovasculature and impaired or missing lymphatic drainage in the tumor.

The potential of the actively and passively targeted polymer probes for the fluorescence-guided surgery of malignant tumors was compared and discussed. Targeted polymeric fluorescent nanoprobes have the potential to increase the safety of oncological surgery in the upper aerodigestive tract and to improve the overall prognosis of patients.

## 2. Materials and Methods

### 2.1. Materials

1-Aminopropan-2-ol, 2,2′-azobis-(isobutyronitrile) (AIBN), *N*,*N*′-diisopropylcarbodiimide (DIC), *N*,*N*-dimethylacetamide (DMA), *N*,*N*-dimethylformamide (DMF), ethyldiisopropylamine (DIPEA), dithiothreitol (DTT), 1-hydroxybenzotriazole (HOBt), methacryloyl chloride, piperidine, trifluoroacetic acid (TFA), triisopropylsilane (TIPS), and all other reagents and solvents were purchased from Sigma-Aldrich (Prague, Czech Republic). TentaGel Rink amide resin, ethyl cyano(hydroxyimino)acetate (Oxyma), (benzotriazol-1-yloxy)-trispyrrolidinophosphoniumhexafluorophosphate (PyBOP), 9-fluorenylmethoxycarbonyl (Fmoc)-amino acid derivatives and 1-(9-fluorenylmethyloxycarbonyl)amino-3,6,9,12,15,18,21,24,27,30,33,36-dodecaoxanonatriacontan-39-oic acid (Fmoc-Peg12-COOH) were purchased from Iris Biotech GmbH (Marktredwitz, Germany). 5-Azidopentanoic acid was obtained from Bachem (Bubendorf, Schwitzerland) and amino-1-(11,12-didehydrodibenzo[b,f]azocin-5(6H)-yl)propan-1-one (Dbco-NH2) was purchased from Click Chemistry Tools (Scottsdale, AZ, USA). Amino and NHS-ester derivatives of fluorescent dyes Cyanine 7 (Cy7-NH2, Cy7-NHS) and Dyomics-633 (Dy-633-NH2, Dy-633-NHS) were obtained from Lumiprobe GmbH (Hannover, Germany) and Dyomics GmbH (Jena, Germany). All amino acids were L-configuration. Human EGF (hEGF) was purchased from Biovision Inc. (Milpitas, CA, USA). Methacryloyl chloride, 1-aminopropan-2-ol, and dichloromethane were distilled immediately before use. All chemicals and solvents were of analytical grade. Solvents were purified and dried using standard procedures.

### 2.2. Physico-Chemical Characterization

Monitoring of the peptide purity and conjugation of the peptide to the reactive copolymer were performed by high-performance liquid chromatography (HPLC) using a Chromolith Performance RP-18e column (100 × 4.6 mm, Merck, Gernsheim, Germany), and a linear gradient of water–acetonitrile, 0%−100% acetonitrile, in the presence of 0.1% TFA with a UV-vis diode array detector (Shimadzu, Kyoto, Japan). Determination of the molecular weight and dispersity of the copolymers was performed by size exclusion chromatography (SEC) on a HPLC system (Shimadzu, Kyoto, Japan) equipped with refractive index, UV, and multiangle light scattering DAWN 8 EOS (Wyatt Technology Corp., Santa Barbara, CA, USA) detector using a TSK 3000 SWXL column (Tosoh Bioscience, Kyoto, Japan) and 80% methanol, 20% 0.3 M acetate buffer pH 6.5 at a flow rate of 0.5 mL/min. The calculation of molecular weights from the light scattering detector was based on the known injected mass assuming 100% mass recovery. The molecular weight and dispersity values of the fluorescently labeled polymers were calculated from SEC chromatograms based on pHPMA calibration, using RI detector data. The content of thiazolidine-2-thione (TT) groups was determined spectrophotometrically on a Helios Alpha UV/vis spectrophotometer (Thermo Fisher Scientific, Waltham, MA, USA) using the molar absorption coefficient for TT in methanol, ε305 = 10,280 L mol^−1^ cm^−1^. Determination of dyes Dy-633 or Cy7 was also determined spectrophotometrically using the molar absorption coefficient for Dy-633 (Cy7) in methanol, ε637 = 159,000 L mol^−1^ cm^−1^ (ε750 = 199,000 L mol^−1^ cm^−1^).

### 2.3. Synthesis of Monomers and Polymer Precursors

Monomers and amino-reactive copolymer precursor poly(HPMA-*co*-Ma-β-Ala-TT) (P-TT) were prepared as described previously [13]. Reactive polymer with dye Dy-633-NH2 or with Cy7-NH2 and control polymers with the dye and without targeting peptides (P-Dy-633 and P-Cy7) were prepared as described previously [13].

### 2.4. Synthesis of Peptide-Targeted Nanoprobes

The peptide-targeted nanoprobes were prepared as described earlier [13] to form the polymer conjugates P-GE11-Dy-633 or P-GE11-Cy7. Similarly, the scrambled control was prepared to form the conjugate P-scrGE11-Cy7 [13]. The amino acid analysis was used for the oligopeptide content determination.

### 2.5. Synthesis of hEGF-Targeted Nanoprobes

HEGF (2 mg) was added to 0.5 mL of 0.03 M phosphate buffer pH 8 to form a suspension and 4 mg of the polymer precursor P-TT-Dy-633 or P-TT-Cy7 was dissolved in 80 µL of DMA. The solution was added dropwise to the suspension of hEGF and vortexed overnight. The next day, the solution was clear, and the completeness of the reaction was verified by HPLC. The reaction mixture was chromatographed on Sephadex G 25 resin in water (PD 10 column, GE Healthcare). The polymeric fraction was freeze-dried. The molecular characteristics of conjugates P-EGF-Dy-633 and P-EGF-Cy7 are shown in Table 1. The amino acid analysis was used for the peptide content determination.

### 2.6. Synthesis of Monoclonal Antibody (mAb)-Targeted Nanoprobe

Radical copolymerization in dimethylsolfoxide (DMSO) was applied for the synthesis of semitelechelic polymer precursor (polymer sP-TT, Table 1) containing Boc-protected hydrazide groups. Functionalized initiator 3,3’-[azobis(4-cyano-4-methyl-1-oxobutane-4,1-diyl)] bis(thiazolidine-2-thione) was employed as described earlier [18]. The end-chain reactive maleimide (MI) group was introduced into copolymer sP-TT by the reaction of TT group with N-(2-aminoethyl)maleimide, as described previously [19].

DY-633 and Cy7 containing polymers, sP-DY-633-MI and sP-Cy7-MI, were prepared from sP-MI by the reactions of the hydrazide groups of the polymeric precursor with NHS-esters of the corresponding dyes. Briefly, the polymer precursor sP-MI (50 mg) and DY-633-NHS ester (2 mg, 2 µmol) were dissolved in 0.5 mL of DMA and the solutions were mixed together. The reaction was monitored by TLC (eluent: methanol:ethyl acetate:acetic acid 10:4:0.5): RfDY-633-NHS ester = 0.84, Rfpolymer = 0. After 2 h at room temperature, TLC showed ca 85% yield and the low-molecular weight impurities were removed by gel filtration (Sephadex LH-20, solvent methanol). The purified polymer probe was isolated by precipitation in ethyl acetate. The mAb targeted nanoprobe was prepared in two consecutive steps as described elsewhere [19]. First, the mAb, cetuximab, was mildly reduced by DTT to introduce sulfanyl groups. Then, the reduced mAb was reacted with semitelechelic copolymers containing MI groups, P-DY-633-MI or P-Cy7-MI, to form the star-like mAb-polymer-dye conjugate. The product was characterized using SEC, UV-VIS spectrophotometry and electrophoresis. The amino acid analysis was used for the mAb content determination.

### 2.7. Synthesis of Star Polymers

The star polymer precursors St-P-1 and St-P-2 were synthesized by grafting the thiazolidine-2-thione (TT)-terminated semitelechelic HPMA copolymer sP-TT onto the 2nd or 4th generation polyamidoamine (PAMAM) dendrimers containing terminal amino groups, as described in [20]. Briefly, semitelechelic copolymer precursor sP-TT with chain-terminated TT groups (32 mg; 0.003 mmol TT groups) was dissolved in 1 mL of methanol and added into a stirring solution of 0.8 mg of D-NH2 (G2, diaminobutane core, 16 amino groups; 0.002 mmol of dendrimer), or 0.4 mg of D-NH2 (G4, diaminobutane core, 64 amino groups; 0.001 mmol of dendrimer), respectively, in 0.4 mL of methanol. After 2 h, the reaction was terminated by adding 5 μL of 1-aminopropan-2-ol and the remaining free amino groups of the dendrimer were end-capped by reaction with acetic anhydride. The star precursor was freed of low molecular weight impurities by gel filtration (Sephadex LH-20, solvent methanol) and subsequently isolated by precipitation in ethyl acetate. TFA was used for the deprotection of Boc-protected hydrazide groups.

Star polymer conjugates St-P-1-Cy7 and St-P-2-Cy7 (Table 1) bearing fluorescent dye Cy7, attached via a hydrazide bond, were prepared by the reaction of star polymer precursors with Cy7-NHS ester, as described above. Reference polymer sP-Cy7 was prepared from the sP-TT precursor after the removal of TT groups with 1-aminopropan-2-ol, deprotection and attachment of Cy7 by the procedure described above.

### 2.8. Cell Culture

The cell lines breast carcinoma MDA-MB-231 and HNSC FaDu were purchased from American Type Culture Collection (ATCC, Manassas, VA, USA). MDA-MB-231 was cultured in a mixture of Dulbecco’s Modified Eagle Medium (DMEM, Gibco, Gaithersburg, MD, USA) and Roswell Park Memorial Institute (RPMI) 1640 medium (1:1, Gibco), supplemented with 10% fetal bovine serum (FBS, Sigma-Aldrich, Prague, Czech Republic) and 1% penicillin–streptomycin (Gibco, Waltham, MA USA). FaDu cells were grown in Minimum Essential Medium Eagle (Sigma-Aldrich, Prague, Czech Republic), enriched with 2 mM L-glutamine (Gibco, Waltham, MA USA), 1% (*v*/*v*) non-essential amino acids (NEAA) solution (Gibco, Waltham, MA USA), 10% FBS (Sigma-Aldrich) and 1% penicillin-streptomycin (Gibco, Waltham, MA USA). Cell lines were maintained at 37 °C in a 5% CO_2_ humidified atmosphere. FaDu cell line, expressing Red Fluorescent Protein (RFP), was a kind gift from Dr. Trzil (BIOPHARM, Research Institute of Biopharmacy and Veterinary Drugs, Liběchov, Czech Republic). All experiments were conducted on cells in passage less than five.

### 2.9. Flow Cytometry

Cells were harvested in 100 mM HEPES buffer with 20 mM NaCl, 10 mM EDTA and 0.5% bovine serum albumin (BSA, Sigma), pH 7.4 (MDA-MB-231) or 0.05% trypsin-EDTA (Gibco; FaDu). Collected cells were centrifuged (1500 rpm, 3 min) and washed once with PBS containing 0.5% BSA (PBS-BSA). After washing, 5 × 10^4^ cells/vial were incubated with the appropriate concentration of tested conjugate or phycoerythrin (PE) conjugated anti-EGFR monoclonal antibody (1:10, Exbio, Prague, Czech Republic) for 1 h at 4 °C in the dark. Afterwards, cells were washed once from unbounded conjugate with PBS-BSA, centrifuged and mixed with Sytox Blue Dead Cell Stain (Thermo Fisher Scientific, Waltham, MA, USA) to distinguish between live and dead cells. Data acquisition was performed using a FACSVerse (Becton Dickinson, Franklin Lakes, NJ, USA) with subsequent analysis using FlowJo software version 10 (Tree Star Inc., Ashland, OR, USA).

### 2.10. Statistical Analysis

All data are presented as mean ± SEM. The values were determined from at least two experiments performed in duplicate. Saturation binding data were analyzed by GraphPad Prism 5 software, ver. 5.1 (GraphPad Software Inc., San Diego, CA, USA) using non-linear regression curve fit (one-site binding model). Kd values were calculated after subtraction of the control conjugate (non-specific binding), as the targeting conjugate concentration occupies 50% of the receptors at equilibrium.

For in vivo experiments, one-way ANOVA and a post hoc procedure consisting of multiple two-sided t-tests with the Holm method to control the family-wise error rate were performed using statistical language R. P values lower than 0.05 were considered statistically significant.

### 2.11. Immunofluorescence Analysis of Cells

Cells in microscopic chambers were fixed with buffered 4% formaldehyde for 15 min and washed in PBS three times. Subsequently, cells were permeabilized by 0.3% Triton X-100 in PBS for 15 min and blocked by 5% BSA in PBS for 30 min. Samples were incubated with primary antibody for 60 min at ambient temperature, then 60 min with secondary antibody at ambient temperature. Finally, 25 µL of Vectashield^®^ antifade medium with DAPI (Vector Laboratories, Inc., 30 Ingold Road, Burlingame, CA 94010, USA) was added to each cell chamber for nuclear staining.

### 2.12. Immunofluorescence Analysis of Tissues

Tumor tissues obtained from the nude mice or human subjects were fixed with buffered 4% formaldehyde for 24 h, dehydrated, embedded in paraffin blocks and cut into 5 µm slides. Subsequently, the tissue slides were deparaffinized, and rehydrated. Antigen retrieval was performed by boiling in sodium citrate buffer (pH 6.0) in a microwave oven for 15 min. Samples were then incubated with primary antibody for 60 min at ambient temperature and with secondary antibody for 60 min. A droplet of Vectashield^®^ antifade medium with DAPI (Vector Laboratories, Inc., 30 Ingold Road, Burlingame, CA, USA) was added per slide for nuclear staining before mounting, as per manufacturer’s protocol.

### 2.13. Antibodies

Primary antibodies anti-EGFR rabbit polyclonal antibody (PA11110, Thermo Fisher Scientific, Pierce Biotechnology, Rockford, IL, USA), dilution 1:200, and anti-human cytokeratin mouse monoclonal antibody AE1/AE3 (M3515, DAKO Denmark A/S, Glostrup, Denmark), dilution 1:200. Secondary antibodies donkey anti-mouse IgG (H+L) highly cross-adsorbed secondary antibody, Alexa Fluor 680 (A10038, Invitrogen, Paisley, UK), dilution 1:1000 and donkey anti-rabbit IgG (H+L) highly cross-adsorbed secondary antibody, Alexa Fluor 488 (A-21206, Invitrogen), dilution 1:1000.

### 2.14. Fluorescence Microscopy

The confocal Leica TCS SP5 microscope (Leica Microsystems, Wetzlar, Germany) was used for imaging. Alexa Fluor 488 was excited using 488 nm argon laser line, Alexa Fluor 680 was excited using 633 nm HeNe laser line. DAPI was excited using 405 nm laser. Open source FIJI ImageJ was utilized for the processing of obtained images [21].

### 2.15. Intravital Tumor Accumulation Assessment

All animal experiments were performed in accordance with the Act on Experimental Work with Animals (Public Notice of the Ministry of Agriculture of the Czech Republic No. 246/1992, No. 311/1997, No. 207/2004; Decree of the Ministry of the Environment of the Czech Republic No. 117/1987; and Act of the Czech National Assembly No. 149/2004) of the Czech Republic, which is fully compatible with the corresponding European Union directives. Animal study protocol, covering all hereafter described animal experiments, was approved by the Czech Ministry of Education Youth and Sports (Approval No. MSMT-12181/2016-4, date of approval: 15 May 2016).

All patients signed informed consent before entering the study and the study protocol was approved by the Central Ethics Committee of Hospital Motol, Prague, Czech Republic (Approval No. AZV16-28594A, date of approval: 24 June 2015). In addition, all data were analyzed with respect to patient privacy.

FaDu-RFP cells, where RFP means red fluorescent protein, were trypsinized, spun and re-suspended in an FBS-free medium at a concentration of 2 × 107 cells per mL. BD Matrigel™ (I.T.A.-Intertact, Ltd., Prague, Czech Republic) was added (1/2 of the cell suspension volume) and 0.15 mL of the mixture was subcutaneously administered (2 × 106 cells per mouse) into the abdominal right flank of athymic nude mice (Velaz, Ltd. and Charles River Laboratories International, Inc., Prague, Czech Republic). When the tumors had reached a minimum size of 6 mm in diameter, mice were divided into groups of eight and randomly allocated for individual intravenous conjugate administration. In case of conjugates sP-Cy7, St-P-1-Cy7 and St-P-2-Cy7, 0.25 mg of probes per mouse were administered, dissolved in 0.1 mL. For the experiment with actively targeted conjugates mAb-P-Cy7, P-scrGE11-Cy7, P-GE11-Cy7 and P-EGF-Cy7, the amount of conjugates was unified based on the Cy7 dye content, as these conjugates had uneven Cy7 content. Tumor accumulation and the biodistribution of polymeric carriers conjugated to Cy7 were determined using the Xtreme In Vivo Imaging System (Bruker BioSpin, Ettlingen, Germany) by intravital imaging 15 min, 4 and 24 h after administration, as described in [13]. Images were taken separately for both RFP and Cy7 fluorescent channel (exposition time of 5 s for both channels), and the reflectance images were acquired. The light source was Xenon Arc Lamp with 400 W power. Band filters were used as excitation emission filters to pass a single wavelength of + - about 5 nm. The camera has a CCD sensor and is cooled to −65 °C. Open source software FIJI was utilized for the adjustment of obtained images and quantification of ROIs [21].

## 3. Results

### 3.1. Synthesis of the Polymer Nanoprobes

Various polymer-based nanoprobes were designed and synthesized, tailoring their potential to visualize tumors to potentiate fluorescence-guided surgery. Linear and star polymer probes and probes targeted with three structurally different EGFR-targeting units, oligopeptide GE-11, EGF and anti-EGFR mAb cetuximab, were successfully prepared as shown in Table 1, using free radical polymerization and controlled RAFT radical polymerization. Using the grafting-to approach, star-like polymer carriers were successfully synthesized by controlled grafting of semitelechelic HPMA copolymers to a PAMAM-based dendrimer core. The molecular weight was controlled by selection of the dendrimer generation and the number of the polymer grafts attached to the dendrimer core, producing polymers with a Mw of 170 and 770 kg/mol, which, with a linear polymer of 30 kg/mol, formed the broad molecular weight sample set.

The GE11- and GE11scr-targeted polymers were prepared by the attachment of oligopeptides to the polymer precursor containing TT groups via an aminolytic reaction. The binding in dry organic solvent led to the complete attachment of oligopeptides; the final nanoprobes contained 15% wt. of the oligopeptide, corresponding to 5–6 oligopeptides attached to each polymer chain. A similar reaction was also employed for the preparation of EGF-targeted nanoprobes in an aqueous buffer at pH 8, which enabled the partial deprotonation of amino groups leading to the aminolytic reaction with TT groups of the polymer precursors. The partial deprotonation was chosen to minimize possible cross-linking resulting from the reaction of multiple amino groups of EGF with polymer precursors. The GPC chromatograms showed that the polymers were not significantly cross-linked, and their molecular weight is equal to the attachment of approximately two hEGF molecules per one polymer chain. Finally, a mAb targeted polymer nanoprobe was synthesized by controlled grafting of the partly reduced mAb molecule, enabling eight polymer carriers attached to one mAb. For the schematic description of the syntheses of peptide-targeted polymer systems, see Scheme 1 and Appendix A.

### 3.2. Binding Affinity of Polymeric Conjugates to EGFR-Positive Cells

To evaluate the binding affinities of prepared polymeric conjugates, first the EGFR expression level was determined in breast cancer cells MDA-MB-231 and hypopharyngeal carcinoma cells FaDu. Flow cytometry analysis revealed that FaDu cells expressed two-times more EGFR than MDA-MB-231 (Figure 1).

Investigation of the targeting ability of prepared conjugates by flow cytometry showed that the mAb-P-Dy-633 conjugate with a Kd of 274 ng/mL (MDA-MB-231) or 591 ng/mL (FaDu) (Figure 2) had the best binding affinity. P-EGF-Dy-633 targeted 50% of the receptors at equilibrium (Kd) at a concentration of 2.33 µg/mL (MDA-MB-231) and 5.24 µg/mL (FaDu). In the case of P-GE11-Dy-633, increasing concentration of the conjugate increased binding affinity to EGFR in MDA-MB-231, but did not reach saturation, even at the highest concentration of 100 μg/mL. In comparison with P-EGF-Dy-633 and mAb-P-Dy-633, the binding affinity of P-GE11-Dy-633 to EGFR in FaDu cells was minimal.

### 3.3. EGFR Expression

As a HNSCC in vivo model, the expression of the target EGFR was evaluated in the human pharyngeal carcinoma cell line FaDu. Expression was determined in vitro cultured cells and mouse xenograft tumor sections. As shown in Figure 3, immunohistochemical staining revealed clear membrane EGFR expression on both FaDu cells and established FaDu tumors. The chosen FaDu HNSCC model is thus appropriate for testing the EGFR-targeted polymeric probes. Moreover, the expression of EGFR by cancer cells in HNSCC tissues was also confirmed (see Figure 3) for 25 independent patient samples. Moreover, the EGFR expression was also validated using Western blot analysis of EGFR expression on FaDu and MDA cells, Appendix A. Indeed, both glycosilated and non-glycosilated variants were found on FaDu cells.

### 3.4. Tumor Accumulation Based on Molecular Weight of Conjugates

To assess the influence of molecular weight on tumor accumulation, three different molecular weights (26, 170 and 770 kg/mol) of conjugates were tested after intravenous application at three time points (15 min, 4 and 24 h after application). In the case of conjugate sP-Cy7 (26 kg/mol), low signal from the tumor site was documented, however, there was no significant increase in the fluorescence signal from the tumor over time (Figure 4). In contrast, higher molecular weight conjugates showed improved tumor accumulation which increased over time. At 4 h after intravenous administration, the accumulation improved significantly for St-P-1-Cy7 (170 kg/mol) and St-P-2-Cy7 (770 kg/mol), with St-P-1-Cy7 being significantly superior to St-P-2-Cy7 (Figure 4).

Similar results were recorded at 24 h. Fluorescent images of whole mice revealed that the lower molecular weight conjugate sP-Cy7 accumulated in kidneys prior to elimination in urine, while the other two conjugates do not show substantial kidney accumulation based on their increased size (Figure 5A). Tumor accumulation was confirmed by colocalization with the RFP signal from the tumor cells (Figure 5B).

### 3.5. Tumor Accumulation Based on Active Targeting

Next, the following conjugates actively targeted against EGFR were tested: samples with clinically approved mAb cetuximab (mAb-P-Cy7), human recombinant EGF (P-EGF-Cy7), GE-11 (P-GE11-Cy7) and sP-Cy7 as a non-targeted control. At 4 and 24 h intervals, the sample with GE-11 (P-GE11-Cy7) exhibited significantly higher accumulation in tumor tissues than in the nontargeted sP-Cy7 conjugate (Figure 6), thus proving the additional value of active oligopeptide targeting.

Figure 7A depicts the distribution over time, revealing the accumulation of samples mAb-P-Cy7, P-scrGE11-Cy7 and P-GE11-Cy7 in the tumor. Efficient tumor accumulation was confirmed by colocalization with the RFP signal derived from the RFP-expressing tumor cells (Figure 7B). In addition to successful tumor accumulation, P-GE11-Cy7 also gave a strong signal from kidneys, indicating concurrent renal excretion (Figure 7).

The conjugate with EGF (P-EGF-Cy7) exhibited no significant accumulation in the tumor over time. At all intervals, the strength of the signal was similar to that of the control polymer (sP-Cy7) (Figure 6). In the case of P-EGF-Cy7, analysis of the Cy-7 signal beyond the tumor area revealed strong accumulation of the conjugate in kidneys and liver (Figure 8A). Additionally, poor tumor accumulation resulted in low colocalization with the tumor-related RFP signal (Figure 8B).

## 4. Discussion

### 4.1. Synthesis of the Polymer Nanoprobes

The free radical and RAFT polymerization techniques were used successfully for polymer nanoprobe synthesis. The RAFT technique resulted in copolymers with a very narrow molecular weight distribution, which exhibit more uniform biodistribution, highly desirable characteristics for a good contrast in the fluorescent imaging. Consequently, this should also facilitate the regulatory approval process for eventual clinical application.

The controlled “grafting-to” approach enabled the tailored synthesis of polymer nanoprobes in a broad range of molecular weights, from 26 to 770 kg/mol. The size of the polymer probes was controlled by selection of the appropriate generation of the PAMAM dendrimer core and by adjustment of the ratio between the polymer and the dendrimer. Similarly, the aminolytic reaction between the reactive TT groups of the polymer precursors and the amino groups of the oligopeptides or EGF under controlled conditions was employed to load a sufficient amount of the EGFR-targeting moieties to the polymer conjugate. For the mAb binding to the polymer, we selected a regio-selective “grafting-to” approach to minimize undesired modification of the binding site of the mAb to EGFR. Thiol groups of mAb, generated upon partial reduction of the disulfide bridges of the protein, were used for a single point attachment of the semitelechelic polymers to mAb.

### 4.2. Binding Affinity of Polymeric Conjugates to EGFR-Positive Cells

The binding affinity of prepared conjugates mAb-P-Dy-633, P-EGF-Dy-633 and P-GE11-Dy-633 was evaluated in vitro using two cell lines, MDA-MB-231 and FaDu. FaDu expressed twice as much EGFR as MDA-MB-231, thus influencing the detected fluorescence intensities. However, the trend in binding affinity was similar in both cell lines, with the highest binding affinity observed for the cetuximab-targeted polymer conjugate mAb-P-Dy-633. The corresponding Kd values for MDA-MB-231 (0.274 µg/mL) and for FaDu cells (0.591 µg/mL) were approximately ten times lower than those observed for P-EGF-Dy-633, (MDA-MB-231 - 2.33 µg/mL; FaDu - 5.24 µg/mL). The results are in agreement with recently published data [22], showing much lower Kd for cetuximab, which can even compete with EGF in binding to EGFR. The two-fold higher calculated Kd for FaDu (in comparison with MDA-MB-231) is in accordance with the two-fold higher expression of EGFR in FaDu cells, Figure 1.

The polymer conjugate P-GE11-Dy-633 showed the lowest binding affinity. Although GE-11 peptide was previously reported as a targeting ligand for the EGFR [23], in our study, its efficiency to bind to EGFR in vitro was the lowest in comparison to both mAb (cetuximab) and the EGF. Indeed, recently it was published that GE-11 containing systems are significantly internalized by the FaDu cells [13], thus we can conclude that, despite their low binding affinity, the GE11-containing polymers could be internalized significantly in a short time. In the case of MDA-MB-231 cells, increasing fluorescent intensity with increasing amounts of P-GE11-Dy-633 was observed, with no saturation up to the peptide concentration of 100 µg/mL; however, the fluorescent intensity of P-EGF-Dy-633 or mAB-P-Dy-633 reached saturation below 10 µg/mL of the protein concentration. Interestingly, when FaDu cells were used, almost no binding affinity of P-GE11-Dy-633 was detected. It is possible that the binding affinity of GE11 to EGFR is variable in different EGFR-positive cell lines, due to the structural variation in EGFR, which does not change the binding affinity of cetuximab and EGF as much as EGFR-binding structures, but is pronounced in the case of GE11, with a much higher Kd for EGFR [24].

### 4.3. Tumor Accumulation Based on Molecular Weight of Conjugates Versus Active Targeting

It seems that nanoprobes with a molecular weight above the renal threshold (50,000 g/mol) and below 500,000 g/mol are optimal for passive targeting to solid tumors [25] due to the EPR effect; while sP-Cy7 is rapidly eliminated via kidney and St-P-2-Cy7 is too big for effective tumor accumulation, St-P-1-Cy7 showed a promising accumulation profile. As expected, mAb-P-Cy7 exhibited pronounced accumulation in the tumor after 15 min, thus showing excellent synergism of passive and active tumor targeting. The conjugated antibody provides a higher molecular weight (above the renal threshold for HPMA copolymers), thus prolonging the circulation time and decreasing renal excretion [26]. Likewise, mAb-P-Cy7 showed significantly superior tumor accumulation at 15 min and 4 h compared with P-GE11-Cy7. At 24 h P-GE11-Cy7 achieved slightly better results, though the difference was insignificant. The outstanding tumor accumulation of P-GE11-Cy7 in vivo is contradictory to the in vitro data presented and discussed above. Indeed, the scrambled GE11-targeted probe did not show any significant difference with respect to the sP-Cy7 accumulation, thus supporting the efficacy of the GE11 oligopeptide for the EGFR targeting in vivo. We hypothesize that passive targeting of P-GE11-Cy7 with a molecular weight close to the limit of the renal filtration is highly elevated and, in combination with the slightly pronounced active EGFR-targeting, leads to the observed tumor accumulation after 24 h. Conversely, the effect of active targeting using mAb is limited by the saturation of the EGFR receptors, as in vitro. The results proved the potential of targeted polymer nanoprobes, demonstrating their advantages and limitations. To achieve rapid tumor accumulation, the best option is highly specific mAb targeting, whereas polymer nanoprobe with targeting oligopeptides, relatively simple in design and synthesis, and with a low cost, is acceptable for navigated surgery when a short time between probe application and surgery is not required. Moreover, the GE11-containing polymer probe is easily clearable from the organism, in contrast to a much larger star or mAb-targeted polymer probes, proving the potential applicability of the GE-11 targeted polymer conjugate for tumor visualization. Moreover, we can also conclude that St-P-1-Cy7 has similar accumulation profile to mAb-P-Cy7, thus proving the prevalence of the EPR effect in the case of high molecular weight systems over active targeting.

Interestingly, P-EGF-Cy7 did not exhibit any significant tumor accumulation, while in vitro results showed a relatively strong interaction with EGFR. However, a strong accumulation of P-EGF-Cy7 in liver and kidney was observed at all time intervals, which may be due to the cross-reactivity of the human EGF [27] with murine EGFR overexpressed in liver [28] and kidney of mice [29]. No significant accumulation of GE11 and mAb cetuximab in liver and kidney was observed, thus proving no or very low cross-reactivity with mouse EGFR. Moreover, we have not observed any sign of toxicity after the injection of the polymer probe, e.g., weight loss or other toxicity symptoms. Similarly, polymer probes did not show toxicity when measured on cancer cell lines in vitro [13].

## 5. Conclusions

Tumor-targeted polymer probes, intended for the visualization of EGFR-positive malignant tumors for successful resection via fluorescence-guided endoscopic surgery, were successfully designed, prepared and evaluated. The optimal molecular weight is 200,000 g/mol for passive tumor targeting, showing maximal accumulation from 15 min to 24 h. The nanoprobes targeted with GE-11 and cetuximab exhibited a very good targeting profile but differed in their tumor accumulation kinetics. Cetuximab should be preferentially used for the visualization of tumors after a short time (15 min after injection), while the GE11-targeted nanoprobe is excellent for visualization of tumors more than 4 h after nanoprobe injection. In conclusion, fluorescent polymer probes targeted with oligopeptide GE11 or with cetuximab are excellent tools for navigation during oncological surgery of HNSCC, due to their relatively simple design, synthesis, low cost, optimal pharmacokinetics and accumulation in tumors.

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
