# Peer review of "Targeted Polymer-Based Probes for Fluorescence Guided Visualization and Potential Surgery of EGFR-Positive Head-and-Neck Tumors"

_pharmaceutics, 2020, doi:10.3390/pharmaceutics12010031_

Round 1

Reviewer 1 Report

In the submitted manuscript, the authors investigated the synthesis and applications of fluorescent probe-labeled polymers for surgery navigation. There are some interesting results, but a number of issues should be addressed.

The authors should provide a synthetic scheme for the preparation of the polymers. This will be quite helpful for the readers to follow the article.  In Table 1, I would suggest the authors provide the Mn values rather than the Mw. The authors synthesized different chain topologies for fluorescent imaging. Will the chain architectures affect the imaging capacity? Which one is the best for fluorescence imaging? After 24 h, the fluorescence can be still observed, suggesting a very slow metabolism process in vivo. How safety is these macromolecular fluorophores? 

Author Response

Responds to Reviewer 1

In the submitted manuscript, the authors investigated the synthesis and applications of fluorescent probe-labeled polymers for surgery navigation. There are some interesting results, but a number of issues should be addressed.

The authors should provide a synthetic scheme for the preparation of the polymers. This will be quite helpful for the readers to follow the article.  In Table 1, I would suggest the authors provide the Mn values rather than the Mw. The authors synthesized different chain topologies for fluorescent imaging. Will the chain architectures affect the imaging capacity? Which one is the best for fluorescence imaging? After 24 h, the fluorescence can be still observed, suggesting a very slow metabolism process in vivo. How safety is these macromolecular fluorophores? 

Answer: We thank the reviewer for his comments. We have added synthetic scheme as Figure A1. Mn values were added into the table 1. We have synthesized and described several polymer structures with the aim to show the influence of the size and type of targeting unit on the potential of the polymer probes for the tumour accumulation and potentially also for navigated surgery. From the study we can conclude that the high-molecular weight polymer have excellent accumulation in the tumour tissue but they circulate for very long time in the organism, thus the overall fluorescence from the body is quite high for long period of time. We believe that the small polymer-GE11 polymer construct is the best candidate for following study, as the polymer is quite small in size, but due to the active targeting the efficacy of tumour accumulation is similar as for the system targeted by monoclonal antibody. Based on the small molecular weight, overall fluorescence of the tissue outside the tumour area would be much lower for the polymer-GE11probe. We have not observed any sign of toxicity of the polymer probes in the range of concentrations used within the study. The polymer carrier is highly water-soluble and biocompatible, non-immunogenic a non-toxic. We have added a sentence regarding the toxicity into the manuscript in part 4.3.

Reviewer 2 Report

The manuscript is about a study on the preparation of a diagnostic tool as a polymer-dye conjugate. A fluorescent dye was conjgated either a linear polymer or that polymer was used to decorate the surface of 1st and 2nd generation PAMAM dendrimers to obtain fluorescently labelled-star shaped polymers. These polymeric architectures were then evaluated for their passive accumulation in tumor tissues in mouse model based on EPR effect. Also, their targeted version has been synthesized to achieve active targeting of tumor. Linear polymer was conjugated to EGFR-binding oligopeptide GE-11 (YHWYGYTPQNVI) and human EGF separately. Lastly, multiple polymer chains were attached to heavy chain of a mAB, anti-EGFR monoclonal antibody cetuximab. Especially GE-11 and cetuximab konjüagted polymers were shown to have the targeting ablility best, but GE-11 konjugated one needed more time to accumulate in the tumor. Introduction part was given enough literature and background about the topic, and obntained data was presented with disccusion clearly.

For a simpler comparision with the drawings presented at Scheme 1, the final cnjugate details in Tablo1 might be highlited, otherwise it is difficult to follow up with the final conjugates.

How was the dye / peptide -conjugation degree calculated on HPMA polymer chains? Or how the number of dye per polymer chain was determined?

3.Conjugates were evaluated at 2 different cell lines: MCF-7 and FaDu, which differ at their EGFR experession levels. In terms of cytotoxicity of these polymers (at least the basic HPMA-Cy7 polymer), cytotoxicity with a healthy cell line could be performed to make sure that the carrier is not toxic to he rest of the body.

If the authors think that the active targeting is due to the bingind to EGFR receptor, the a cellular internalization experiments could be done in the presence of free EGFR antibody or pre-treated cell line, and the results for the uptake efficiency of the above-mentioned cell line would give a better idea.

Did authors have chance to check biodistribution of these conjugates, the organ accumulation in a healthy animal. Or analyzed the organ accumulation in at least tumor-bearing aminlas?

Did authors notice any difference in clearance rate of these conjugates, espeally comparing the linear vs star-shaped polymeric construct, since we know that multiarm architecture effects the clearance rate of the macromolecules and let them circulate in the body longer time.

Author Response

Responds to Reviewer 2

The manuscript is about a study on the preparation of a diagnostic tool as a polymer-dye conjugate. A fluorescent dye was conjgated either a linear polymer or that polymer was used to decorate the surface of 1st and 2nd generation PAMAM dendrimers to obtain fluorescently labelled-star shaped polymers. These polymeric architectures were then evaluated for their passive accumulation in tumor tissues in mouse model based on EPR effect. Also, their targeted version has been synthesized to achieve active targeting of tumor. Linear polymer was conjugated to EGFR-binding oligopeptide GE-11 (YHWYGYTPQNVI) and human EGF separately. Lastly, multiple polymer chains were attached to heavy chain of a mAB, anti-EGFR monoclonal antibody cetuximab. Especially GE-11 and cetuximab konjüagted polymers were shown to have the targeting ablility best, but GE-11 konjugated one needed more time to accumulate in the tumor. Introduction part was given enough literature and background about the topic, and obntained data was presented with disccusion clearly.

For a simpler comparision with the drawings presented at Scheme 1, the final cnjugate details in Table1 might be highlited, otherwise it is difficult to follow up with the final conjugates.

Answer: We have changed the Table 1 with the aim to increase the readability of the manuscript. Polymer precursors are now listed in upper part of the Table 1 and the final polymer conjugates in the bottom part.

How was the dye / peptide -conjugation degree calculated on HPMA polymer chains? Or how the number of dye per polymer chain was determined?

Answer: The dye content was determined as mentioned in part 2.2. using the spectrophotometry. The final content was calculated as wt% of the dye in the conjugate. The content of peptide or oligopeptide was measured using amino acid analysis and calculates as wt% of the peptide in the conjugate. We have add the information into the part 2.4. and 2.5.

Conjugates were evaluated at 2 different cell lines: MCF-7 and FaDu, which differ at their EGFR experession levels. In terms of cytotoxicity of these polymers (at least the basic HPMA-Cy7 polymer), cytotoxicity with a healthy cell line could be performed to make sure that the carrier is not toxic to he rest of the body.

 Answer: We thank the reviewer for this comment. These experiments were carried out recently and published, Pola et.al. Multifunct. Mater. 2019, 2, 24004. We have not found toxicity up to the very high concentration of the polymer-dye conjugates. We have added the reference to the manuscript and mentioned the toxicity on these cells lines in part 4.3.

If the authors think that the active targeting is due to the bingind to EGFR receptor, the a cellular internalization experiments could be done in the presence of free EGFR antibody or pre-treated cell line, and the results for the uptake efficiency of the above-mentioned cell line would give a better idea.

Answer: We appreciate the comment of the reviewer as this is really interesting question. Nevertheless, we believe that these types of experiments are possible to provide with valuable results in the case, when all tested peptides or antibodies recognize the same part of the epitope as EGRF antibody. In general, the EGFR possess from four main domain, could be found in the non-glycosylated or glycosylated form and could be presented also as dimer. Based on the preliminary results, we are convinced that it could be hard to completely compete using one EGFR antibody the specific site for all of EGF, Erbitux and GE11, because each of them bound to different sites on EGFR predominantly. Thus we have not included such data into the manuscript.

Did authors have chance to check biodistribution of these conjugates, the organ accumulation in a healthy animal. Or analyzed the organ accumulation in at least tumor-bearing aminlas?

Answer: As the present study was not dealing with the determination of the whole body we have not check the biodistribution in all the organs, but we focused onto the tumor accumulation and comparison of various polymer-based structures for the targeting. Nevertheless, several papers were published recently showing the biodistribution of similar polymers, linear, star or olipopeptide targeted, e.g.           Hofman S.,etal, Biomacromolecules 13 (2012) 652-663; 69.            Petr Chytil, etal J. Controlled Release 172 (2013) 504 – 512; Etrych T., , J. Controlled Release, 164 (2012) 346 – 354. Thus we believe that the body biodistribution is exceeding the scope of the paper.       

Did authors notice any difference in clearance rate of these conjugates, espeally comparing the linear vs star-shaped polymeric construct, since we know that multiarm architecture effects the clearance rate of the macromolecules and let them circulate in the body longer time.

Answer: We thank the reviewer for this interesting comment. Yes, we have determined the difference in the clearance rate for presented polymers recently and we have found that with the increasing size and molecular weight these polymer circulate for much more longer time. Even the limit of the renal threshold is differing for both, linear and star polymers. Even linear polymers with 70 000 g/mol molecular weight can go through the kidney by worm-like motion. In the case of star polymers the limit is somehow smaller, let say around 50 000 g/mol. The data regarding different biodistribution and clearance rate could be found elsewhere (Etrych T., et al Biodegradable star HPMA polymer-drug conjugates: biodegradability, distribution and anti-tumour efficacy, J. Controlled Release 154, 241-248 (2011); Etrych T., et al., Fate of HPMA-based polymer-doxorubicin conjugates in organism: effect of molecular weight and architecture of polymer carrier, J. Controlled Release, 164 (2012) 346 – 354; Petr Chytil, et al., Dual Fluorescent HPMA Copolymers for Passive Tumor Targeting with pH-Sensitive Drug Release II: Impact of Release Rate on Biodistribution, J. Controlled Release 172 (2013) 504 – 512). As the GE11 targeted polymer probe is similarly efficient in the tumor accumulation as the star polymer probes, we believe that GE11-targeted probe is better from the clearance-point of view. We have added such information into the manuscript, part 4.3.    

Reviewer 3 Report

In this work Pola et al. report the design, synthesis and evaluation of HMPA-conjugated fluorescent probes targeting EGFR-positive tumor to support tumor resection via fluorescent-guided endoscopic surgery. The manuscript is well written, and the experimental data well presented.

Indeed, a tool able to highlight tumor margins and to guide tumor resection could have an important impact on patient treatment and improvement of therapy outcome. Despite this, as evidenced by the authors in the discussion and in the conclusions, the in vivo and in vitro results are extremely contradictory.

Follow my concerns about the utility of the GE11-based probes.

The use of drug loaded nanoparticles (NPs) is nowadays a commonly accepted methodology that results in homing a high concentration of drugs to the tumor via a balance between passive penetration due to EPR effect and specific tumor cell targeting. In this regards, the authors show convincing preliminary in vitro immunohistochemical staining on tumor cell membrane. The signal associated to the membranes is described as result of the specific binding of their probes.

The results reported for EGF-decorated NPs and p-GE11-Cy7 probe show a different pattern of cell binding / tumor accumulation. Due to the fact, that for GE11 peptide an alternative EGFR signaling pathway has been described, compared to that induced by EGF interaction with EGFRs.

Did the authors investigate the rate of intracellular penetration of these nanoparticles, in particular for the p-GE11-Cy7 probe using the selected cell lines?

The result of these experiments could help in explaining other data reported in the manuscript.

Could a slow rate of intracellular penetration of the p-GE11-Cy7 nanoparticles be responsible for the low tumor accumulation measured at 4h p.i.? This point should be discussed.

A probe based on the scrambled version of GE11 was tested. Please introduce it in the text, and report the reference Pola et al. 2019.

Pag 9 line 307 and Fig 2: as stated in the text and showed in the figure, for P-GE11-Dy-633 saturation is not reached until a concentration of 100 µg/mL. Despite this, kD values are presented. Please, report if a method different than the saturation binding assay was used to meausure the KD.

Pag11 Fig 5B: please report in the figure legend the time point the images refer to. A scale bar for Cy7 in green and a scale bar for RFP need to be added.

Pag 12 Figure 6: Since they possess a similar size, a comparison between sP-Cy7 and P-GE11-Cy7 is also possible. Differently, in the case of mAb-P-Cy7 also St-P-1 must be considered for a comparison. Please report also the results for the P-scrGE11-Cy7 compound, since it was also tested in vivo.

14 lane 415: Polymeric nanoparticles decorated with the GE11 peptide for active targeting of the epidermal growth factor receptor (EGFR) demonstrated very low to absent affinity binding to EGFR expressed on the surface of selected tumor cell lines. It is difficult to interpret these in vitro results, in particular if no binding on FaDu cells (high EGRF expression) and MDA-MB-231 cells (low EGFR expression) is reported. The following questions rise:

How many times has been the FACS analysis performed? Please report clearly in the text this information for all measurements involving tumor cell lines! Have the authors confirmed the expression of EGFR obtained by FACS analysis for both cell lines via PCR? To what the author refer when talking about affinity for EGFR and dissociation constant? Why the GE11-decorated nanoparticles do not accumulate to the high EGFR expressing tumor xenografts (FaDu) and do bind to the low EGFR expressing (MDA-MB-213) xenografts? Can the authors explain why? Is it due to the probe or is it an effect of the cells biology?

The authors provide as possible explanation for this result the structural variation of EGFR induced by binding with GE11 peptide.

EGFR variants have been described for defined tumor reflecting the presence of distinct subclonal population within the tumor, i.e. glioblastoma, but not between tumors. In this case, the results are even strange, because low or no binding of the GE11-decorated nanoparticles was measured on the high EGFR expressing cell line.

Therefore, the explanation provided is not convincing.

A GE11-derived conjugate namely [64Cu]Cu-NOTA-linker-β-Ala-GE11 has been described as probes for imaging EGFR tumors. For this probe no specific binding in vitro and low tumor uptake in FaDu tumor-bearing mice has been reported. These results suggested that the GE11 conjugates are not suitable for biological investigations, since they presumably form aggregates in vivo.

Did the authors investigated if any aggregation of the probe occurred?

The visualization of the tumor by visualizing the RFP-expressing tumor cells is reported.

There is not mention in the material & method section about the transfection of the cell lines used for expressing RFP. Please report clearly in the M&M section.

14 lane 417: it is reported ….”In the case of MDA-MB-231 cells, increasing fluorescent intensity with increasing amount of P-GE11-418 Dy-633 was observed, with no saturation up to the polymer concentration of 100 μg/mL”….

The fact that no saturation effect has been measured for p-GE11-Cy7 probe is an indication that accumulation occurs only via EPR effect.    

In principle, these results suggest a non-specific accumulation into tumor of the probes, due only to the big size of the NPs. If this is the case, the presence of the GE11 peptide on the surface of the NPs is not mandatory.

Based on these contradictory data, I don’t see any application for the p-GE11-Cy7 probe, in particular as specific agent for targeting EGFR.

Author Response

Responds to Reviewer 3

In this work Pola et al. report the design, synthesis and evaluation of HMPA-conjugated fluorescent probes targeting EGFR-positive tumor to support tumor resection via fluorescent-guided endoscopic surgery. The manuscript is well written, and the experimental data well presented. Indeed, a tool able to highlight tumor margins and to guide tumor resection could have an important impact on patient treatment and improvement of therapy outcome. Despite this, as evidenced by the authors in the discussion and in the conclusions, the in vivo and in vitro results are extremely contradictory. Follow my concerns about the utility of the GE11-based probes.

The use of drug loaded nanoparticles (NPs) is nowadays a commonly accepted methodology that results in homing a high concentration of drugs to the tumor via a balance between passive penetration due to EPR effect and specific tumor cell targeting. In this regards, the authors show convincing preliminary in vitro immunohistochemical staining on tumor cell membrane. The signal associated to the membranes is described as result of the specific binding of their probes.

 The results reported for EGF-decorated NPs and p-GE11-Cy7 probe show a different pattern of cell binding / tumor accumulation. Due to the fact, that for GE11 peptide an alternative EGFR signaling pathway has been described, compared to that induced by EGF interaction with EGFRs. 

Did the authors investigate the rate of intracellular penetration of these nanoparticles, in particular for the p-GE11-Cy7 probe using the selected cell lines?  The result of these experiments could help in explaining other data reported in the manuscript.

Answer: The intracellular penetration of described polymer probes was studied recently and published in Pola et al. Multifunct. Mater. 2 (2019) 024004. We have added the text dealing with the intracellular penetration into the manuscript in part 4.2.

Could a slow rate of intracellular penetration of the p-GE11-Cy7 nanoparticles be responsible for the low tumor accumulation measured at 4h p.i.? This point should be discussed.

 Answer: We do not agree fully with the comment of the reviewer. The accumulation of GE11-targeted polymer is after 4 h much higher than of the nontargeted polymer with similar molecular weight. The accumulation is almost similar to that observed for the mAb-targeted or star system. Thus we believe that the results obtained with GE-targeted system is, with respect to its lower molar mass, positive and show the potential of such probe for the tumor visualisation

A probe based on the scrambled version of GE11 was tested. Please introduce it in the text, and report the reference Pola et al. 2019.

 Answer: The synthesis of scrambled version was mentioned in the M&M part of the manuscript and the reference to Pola 2019 was mentioned.

Pag 9 line 307 and Fig 2: as stated in the text and showed in the figure, for P-GE11-Dy-633 saturation is not reached until a concentration of 100 µg/mL. Despite this, kD values are presented. Please, report if a method different than the saturation binding assay was used to meausure the KD.

Answer: We thank the reviewer for pointing out this inaccuracy and the respective sentence was corrected in the part 4.2. No KD value for the P-GE11-Dy-633 was found. We have employed no other method than the saturation binding assay.

Pag11 Fig 5B: please report in the figure legend the time point the images refer to. A scale bar for Cy7 in green and a scale bar for RFP need to be added.

 Answer: We have added the time point information in the legend. The Figure 5B and 7B is added to illustrate the co-localization of the signal from the RFP signal coming from the tumour cells and the Cy7 signal the coming from the polymer, thus we believe that there is no need to show the scale bar for both signal. Even more, the scale bar is presented for the left part of Figures showing the Cy7 signal.

Pag 12 Figure 6: Since they possess a similar size, a comparison between sP-Cy7 and P-GE11-Cy7 is also possible. Differently, in the case of mAb-P-Cy7 also St-P-1 must be considered for a comparison. Please report also the results for the P-scrGE11-Cy7 compound, since it was also tested in vivo.

 Answer: We appreciate the comment of the reviewer. Nevertheless, sP-Cy7 and P-GE11-Cy7 was compared in the original version in the Figure 6 as the sp-Cy7 is the reference polymer for the GE11 targeted probe. Definitely, we agree that the star a monoclonal antibody targeted systems should be compared and thus we have added this comparison into the manuscript into the part 4.3. Similarly, we have added the sentence dealing with the results with scrambled GE11 oligopeptide to the same part of the manuscript.

14 lane 415: Polymeric nanoparticles decorated with the GE11 peptide for active targeting of the epidermal growth factor receptor (EGFR) demonstrated very low to absent affinity binding to EGFR expressed on the surface of selected tumor cell lines. It is difficult to interpret these in vitro results, in particular if no binding on FaDu cells (high EGRF expression) and MDA-MB-231 cells (low EGFR expression) is reported. The following questions rise:

 How many times has been the FACS analysis performed? Please report clearly in the text this information for all measurements involving tumor cell lines! Have the authors confirmed the expression of EGFR obtained by FACS analysis for both cell lines via PCR? To what the author refer when talking about affinity for EGFR and dissociation constant? Why the GE11-decorated nanoparticles do not accumulate to the high EGFR expressing tumor xenografts (FaDu) and do bind to the low EGFR expressing (MDA-MB-213) xenografts? Can the authors explain why? Is it due to the probe or is it an effect of the cells biology?

The authors provide as possible explanation for this result the structural variation of EGFR induced by binding with GE11 peptide.

 Answer: We thank the reviewer for interesting comment. All the FACS analyses were performed as two experiments performed in duplicate as mentioned in Fig.2 legend and in M&M. The Figure 2 FaDu was rescaled to better visualize the binding MFI for the EGF- and GE-targeted polymers. As it can be shown from rescaled graph, the binding to FaDu is for the GE11-targeted probe almost the same as in the case of MDA-MB-231 cells. We hypothesize that the binding activity is not increased as in the case of the mAb or EGF because of the cells biology. The experiments which can explain this observation are under way, but we are convinced that they go beyond this article. Expression of EGFR was validated also by Western blotting and the results is attached as Figure A2 in the manuscript

EGFR variants have been described for defined tumor reflecting the presence of distinct subclonal population within the tumor, i.e. glioblastoma, but not between tumors. In this case, the results are even strange, because low or no binding of the GE11-decorated nanoparticles was measured on the high EGFR expressing cell line.

Therefore, the explanation provided is not convincing.

 Answer: We appreciate the comment of the reviewer. Several paper could be found investigating the hetegrnicity in the EGFR variants also between the tumors, see Kohsaka S., etal, Tumor clonality and resistance mechanisms in EGFR mutation-positive non-small-cell lung cancer: implications for therapeutic sequencing Future Oncol. 2019 Feb;15(6):637-652. doi: 10.2217/fon-2018-0736. Moreover Western Blotting analysis, see Figuer A2 in the manuscript, is showing that glycosylated and also non-glycosylated variant of the EGFR is present in FaDu cells. Thus we believe that the low in vitro determined binding efficacy is base manly on the structural variation of EGFR, which do not change the binding affinity of cetuximab and EGF.

.

GE11-derived conjugate namely [64Cu]Cu-NOTA-linker-β-Ala-GE11 has been described as probes for imaging EGFR tumors. For this probe no specific binding in vitro and low tumor uptake in FaDu tumor-bearing mice has been reported. These results suggested that the GE11 conjugates are not suitable for biological investigations, since they presumably form aggregates in vivo.

Did the authors investigated if any aggregation of the probe occurred?

  Answer: We have deeply investigated the solution behaviour of the polymer systems. We have not observed any aggregation. We are convinced that the overall properties of GE11 targeted polymer probe are influenced mainly by the properties of the biocompatible and fully water-soluble polymer material. The polymer probe contained only 15wt% of the peptide and thus the overall physico-chemical properties of the system is driven by the polymer material.

The visualization of the tumor by visualizing the RFP-expressing tumor cells is reported.

There is not mention in the material & method section about the transfection of the cell lines used for expressing RFP. Please report clearly in the M&M section.

  Answer: We have added a sentence regarding the RFP-expressing tumour cells into the M&M.

14 lane 417: it is reported ….”In the case of MDA-MB-231 cells, increasing fluorescent intensity with increasing amount of P-GE11-418 Dy-633 was observed, with no saturation up to the polymer concentration of 100 μg/mL”….

The fact that no saturation effect has been measured for p-GE11-Cy7 probe is an indication that accumulation occurs only via EPR effect.    

  Answer: We thank the reviewer for this interesting comment. Nevertheless, when the result of P-Cy7 is compared with that of GE11-targeted, P-GE11-Cy7, one can see significant difference which cannot be escribed to the EPR effect as the size and molecular weight of both polymers is similar. We are convinced that the superior accumulation of GE -11 targeted polymers is based on the synergism of the passive and active targeting to the EGFR positive tumour.

In principle, these results suggest a non-specific accumulation into tumor of the probes, due only to the big size of the NPs. If this is the case, the presence of the GE11 peptide on the surface of the NPs is not mandatory.

Based on these contradictory data, I don’t see any application for the p-GE11-Cy7 probe, in particular as specific agent for targeting EGFR.

 Answer: Based on the respond mentioned above we disagree with the comment of the reviewer. In the manuscript we have proved that the GE-11 targeted polymer is significantly better in the tumour accumulation in comparison to the similar nontargeted polymer system, which is similar in size. Even more, we have proved that the GE11-targeted system has almost the same tumour visualization capacity as mAb-targeted system or star system, which show much more pronounced EPR effect. This mean that the active targeting using GE11 with small EPR effect based on the Mw off 45000 g/mol can be equal to the highly performed EPR effect of star polymer system. Previously, almost 5-10 higher accumulation of such star polymers when compared to linear polymers was described, Etrych T, Journal of controlled Release, 2011, 154, 241-248 (2011). Thus we believe that the results in the manuscript are showing the potential of GE11-targeted probes for tumour visualization.

Reviewer 4 Report

In this manuscript, the authors developed polymer-based probes to detect the EGFR-positive tumors, and explore the molecular weights and imaging windows of the nanoprobes for the improvement of tumor detection. While the clear identification of tumor by the designed polymers on animal model is the strength, the conclusion of this study is weakened by several important factors.

Line 225: this study includes both animal and human tumor tissue. However, the authors only mentioned their experiments were “in accordance with the Act on Experimental Work with Animals” (Line 246). The authors should disclose the approved animal protocol numbers. In addition, there was no information on human study protocol number, how the human tissues were collected, and how the relevant experiments were carried out. The title should be more specific. This study was mainly a nice animal cancer model study. Plus, there were no “endoscopic surgery” component in the study. The authors should specify the animal model and delete “endoscopic” in the title. Compared to the many other reported synthesized receptor-targeted dyes, what are the pros and cons of your synthesized fluorescent polymers? For the animal experiments, the authors should specify the numbers of the nude mice used in each figure, e.g. Fig. 4 - Fig. 8. The authors claimed some fabricated fluorescent polymers passively accumulated at tumors by EPR effect. EPR effect is heavily dependent on the polymer size. What is the size of your fabricated polymer? What about the toxicity, e.g. LD50, of your synthesized dyes? Due to the dramatical metabolism between mice and human, how do you expect the conclusions are translatable to the human study? The authors should always explain the full names of abbreviations when they are first used in the manuscript, such as HPLC in Line 106; DMSO in Line 140; PAMAM in Line 161, RFP in 251, and so on. Comparing Fig. 5 and Fig. 7, EGFR did not improve (sometime even made worse contrast) the dye accumulation at the tumor site, how did the authors explain this effect? Was the EGFR-binding site not effectively conjugated during the synthesis? Did the authors test the conjugation efficiency after each step of synthesis? Line 263: exposition time? Or exposure time? In addition, 5s of exposure time is not compatible with in vivo intraoperative imaging in the human clinical setup. In addition, little information was given to the rest of imaging system, such as light source power. More citations should be given in the Introduction section, especially in the first paragraph of Page 1. All the “in vitro” and “in vivo” should be formatted as the Italic style.

Author Response

Responds to Reviewer 4

In this manuscript, the authors developed polymer-based probes to detect the EGFR-positive tumors, and explore the molecular weights and imaging windows of the nanoprobes for the improvement of tumor detection. While the clear identification of tumor by the designed polymers on animal model is the strength, the conclusion of this study is weakened by several important factors.

Line 225: this study includes both animal and human tumor tissue. However, the authors only mentioned their experiments were “in accordance with the Act on Experimental Work with Animals” (Line 246). The authors should disclose the approved animal protocol numbers. In addition, there was no information on human study protocol number, how the human tissues were collected, and how the relevant experiments were carried out.

Answer: We have added the required information into the M&M section, see 2.15. section.

 The title should be more specific. This study was mainly a nice animal cancer model study. Plus, there were no “endoscopic surgery” component in the study. The authors should specify the animal model and delete “endoscopic” in the title. Compared to the many other reported synthesized receptor-targeted dyes, what are the pros and cons of your synthesized fluorescent polymers?

Answer: We thank the reviewer for the comment. We have changed the title of the manuscript accordingly, “endoscopic” was removed and words “Visualization and Potential” and “Head-and-Neck” were added into the title. Presented HPMA polymer probes have excellent properties, as biocompatibility, non-immunogenicity, non-toxicity and water-solubility, which make the excellent candidates for the medicinal use. These polymers were used in preclinical and clinical studies, K. Ulbrich, et al., Targeted Drug Delivery with Polymers and Magnetic Nanoparticles: Covalent and Noncovalent Approaches, Release Control, and Clinical Studies, Chem. Rev., 116 (2016). Even more these polymer probes could combine passive and active tumour accumulation thus being very good candidates for further study. The structure of the polymers and their synthetic procedure enable easy variation of molecular weight, content and type of the active molecules. Thus we believe that these polymer probes are worth of investigation for fluorescence guided surgery.

For the animal experiments, the authors should specify the numbers of the nude mice used in each figure, e.g. Fig. 4 - Fig. 8. The authors claimed some fabricated fluorescent polymers passively accumulated at tumors by EPR effect. EPR effect is heavily dependent on the polymer size. What is the size of your fabricated polymer? What about the toxicity, e.g. LD50, of your synthesized dyes? Due to the dramatical metabolism between mice and human, how do you expect the conclusions are translatable to the human study?

Answer: The number of animals was specified in the Figures as mentioned by the reviewer. The size of the polymers is listed in Table 1. The molecular weights are from 35-45 000 g/mol for linear polymers up to the 450 000 g/mol star and mAb-containing polymers. Based on the different molecular weights of used systems, there is also different EPR effect of the polymers. Nevertheless, we have proved that the combination of EPR effect and active targeting is highly beneficial for GE11-targeted probes. We have not find any sign of the toxicity of the polymer probes used in the concentrations needed for the fluorescent imaging. The LD50 was not determined as even 100x higher dose of the polymer was not equal to maximal tolerated dose, weight of the animals is reduce more than for 15%. The potential toxicity could be afterward escribe more to the way of administration of huge amount of polymer material. As mentioned above for the study was used the concentration which is more than 100x times lower that the concentration showing some sign of toxicity. Toxicity issue is mentioned in the paper in the part 4.3.We thank the reviewer especially for the last part of the comment. As published elsewhere similar polymers based on the similar polymer material were also potent in humans, H. Dozono, et al., HPMA copolymer-conjugated pirarubicin in multimodal treatment of a patient with stage IV prostate cancer and extensive lung and bone metastases, Targeted Oncology 11 (2016)  101-106. Thus we believe that the polymer probes could be efficient also in humans for visualisation of not-deeply located tumours, e.g. head-and-neck, breast, etc.    

The authors should always explain the full names of abbreviations when they are first used in the manuscript, such as HPLC in Line 106; DMSO in Line 140; PAMAM in Line 161, RFP in 251, and so on.

Answer: All the abbreviation were corrected and explained, when first used.

Comparing Fig. 5 and Fig. 7, EGFR did not improve (sometime even made worse contrast) the dye accumulation at the tumor site, how did the authors explain this effect? Was the EGFR-binding site not effectively conjugated during the synthesis? Did the authors test the conjugation efficiency after each step of synthesis?

Answer: We thank the reviewer for this interesting comment. On these figures various polymers differing in the structure and accumulation mechanism are compared. While Figure 5 is dealing solely with the EPR effect and the best sample is the star system, Figure 7 is dealing with the active targeting. The appropriate control for the GE11-targeted system is the P-Cy7 polymer, which have the same molecular weight and it is clearly visible that the GE11-targeted system in much more better than the non-targeted. Thus the benefit of the EGFR targeting was really proved. The star polymer is accumulated significantly in the tumour just on the basis of its high molecular weight. Thus, we do not think that we can directly compare the results of the star polymer with linear GE11-containing polymer, which both differ extremely in the size. We believe that star polymer decorated with the GE11 oligopeptide will be even more effective than the pure star polymer. We have not tested the efficacy of the polymer systems during the synthesis, but the binding activity was evaluated with the polymers prepared for the biological study.   

Line 263: exposition time? Or exposure time? In addition, 5s of exposure time is not compatible with in vivo intraoperative imaging in the human clinical setup. In addition, little information was given to the rest of imaging system, such as light source power. More citations should be given in the Introduction section, especially in the first paragraph of Page 1. All the “in vitro” and “in vivo” should be formatted as the Italic style.

Answer: We believe that the exposure time is the correct wording. We agree with the reviewer that the in vivo human surgery would need to have another setting, but in the case of mouse biodistribution we have used this setting to observe clearly the signal from the tumour tissue. During the surgery different setup would be used with the aim to enable the surgery in the real time. Clear description of the imaging system was added into the part 2.15. We have added several citations in the introduction as mentioned by the reviewer. All the in vivo and in vitro was transformed to italic style.

Round 2

Reviewer 1 Report

The reviewer's questions have been addressed and the paper can be accepted for publication.

Reviewer 3 Report

The authors have clearly answered to all comments, and clarified some points.